# Experimental Determination of Mechanical Properties of Waste Tyre Bales Used for Geotechnical Applications

**DOI:** 10.3390/ma14123310

**Published:** 2021-06-15

**Authors:** Aleksander Duda, Tomasz Siwowski

**Affiliations:** Department of Roads and Bridges, Faculty of Civil and Environmental Engineering and Architecture, Rzeszow University of Technology, 35-959 Rzeszow, Poland; aduda@prz.edu.pl

**Keywords:** waste tyres, tyre bales, mechanical properties, stiffness, shear strength, creep

## Abstract

Waste tyre-derived products (TDP) are used in some engineering applications and thereby reduce the potential impact on the environment, for example, as lightweight materials in geotechnical engineering projects. One of TDPs is the baling of whole waste tyres to produce rectilinear, lightweight, permeable bales of high bale-to-bale or bale-to-soil friction. The use of lightweight tyre bales in road construction has the potential to satisfy the demand for low-cost materials exhibiting such a beneficial property. This paper presents a laboratory study on the mechanical properties of tyre bales. The laboratory tests included measurement and evaluation of full-scale tyre bales to determine basic values for the geometry and unit weight, compressibility characteristics of tyre bales, including Young’s modulus and Poisson ratio, shear strength along the tyre–tyre and tyre–soil surfaces, creep and stiffness degradation under cyclic load. The respective test procedures and results of these tests are presented in the paper. The paper provides the mechanical properties of tyre bales required for geotechnical projects, as follows: the unit weight—0.515 Mg/m^3^, the Young’s modulus—826 kPa, the Poison’s ratio—0.11, the dry tyre–tyre interface: cohesion of 0.03 kPa and friction angle of 46.0°, the wet tyre–soil interface: cohesion 0.77 kPa and a friction angle of 29.6°, creep deformation of 6.1% of the average height of the bale, and no stiffness degradation of tyre bales under cyclic load. These results could be directly applied for the designing and construction of the tyre-baled structures.

## 1. Introduction

Since 2000 in Europe the EU Landfill Directive [1] has forbidden the disposal of waste tyres in a landfill. Since then waste tyre-derived products (TDP), including whole tyres, tyre bales, shreds, chips, and crumb rubber, have been widely used also in geotechnical applications. The majority of these applications has addressed tyre-derived aggregate (TDA), i.e., shreds, chips, and crumb rubber, for use in road works [2,3,4,5,6]. An alternative is the baling of whole waste tyres to produce cuboidal, lightweight, permeable bales. Typical waste tyre bales comprise 100 to 115 car tyres compressed into a block and secured by galvanized steel tie wires running around the length and depth of the bale. The dimensions of the block are approximately 1.30 m × 1.50 m × 0.80 m and the blocks have a density of around 0.55 Mg/m^3^ [7]. Waste tyre bales have considerable potential for use in geotechnical applications, particularly where their low density, permeability, and ease of handling give them an advantage. The use of lightweight bales in the construction of a lightweight embankment or road foundation over soft ground, the backfilling of retaining structures, the slope stabilization or landslide repairs has the potential to satisfy the demand for low-cost materials exhibiting such a beneficial property.

Tyre bales were first used in the late 1990s in the USA. Since then, over 30 geotechnical projects have been implemented, mainly in the USA and UK. In 2000, in Chautauqua County, NY, USA, a total of five projects were implemented, involving the use of lightweight tyre bales as a subgrade replacement for roads over soft ground [8]. A similar example of the use of tyre bales as a foundation on weak soil is the section of the B-871 road in Sutherland, Scotland, opened in 2003 [9]. Successful applications involving the construction of tyre bale road foundations have been also achieved in the UK in 2005 [10,11]. The remediation of soil cut slope instability in Tarrant County, TX, USA, was achieved in 2002 using tyre bales as a backfill material [12]. Another example of tyre bales being used to rebuild an escarp is the retrofit of a dyke section on the Witham River near Lincoln, UK [13]. Winter et al. [14] describe the key features of tyre bales used for slope failure repair. The most spectacular example of the use of tyre bales in the embankment structure is the section of the A-421 motorway in the UK in 2010 [15]. On the critical section of the road, there were low-bearing soft-plastic soils with a thickness of approximately 20 m. Instead of strengthening the subgrade with 26 m long piles, it was decided to use light filler material. After the economic analysis, the tyre bales were selected and built-in as the material fulfils all the requirements. A case study of the tyre bales use in the construction of a UK flood embankment is presented by Bo and Yarde [16]. The pilot project scheme has used around 1 million tyres, which constitutes approximately 10% of the total number of tyres that would normally go to landfill in any one year in the UK. Tyre bales were also used as modular blocks to build gravity-retaining walls. Several such retaining walls were built in New Mexico, USA [17]. Currently, the first worldwide application of tyre bales for bridge abutment backfilling has been realized in Poland [18]. 

Proper design of tyre bale structures requires a reliable stability analysis to ensure an adequate factor of safety. The stability analyses must utilize the mechanical properties of the tyre bales, which must be properly determined. To date, the authors know only a few US research which has been caried out to determine the mechanical properties of the tyre bales. The first research work was conducted at the University of Texas at Austin. La Rocque [19] and Zornberg & La Rocque [20] summarize the results of the research works conducted to determine the mechanical properties of tyre bales for use in stability analyses, such as unit weight, stiffness, interface shear strength, expansive pressure, and time-dependent deformations under sustained load. Following these works, the next report [21] includes the results of laboratory tests on tyre bales to provide the required information on bale geometry, both air dry and submerged unit weights, permeability, unconfined and confined compressibility characteristics, creep behavior, and shear strength between the two tyre bales. Again at the University of Texas, a laboratory testing program was conducted to determine the compressibility properties of the bales required for design [22]. Finally, Winter et al. [8] carried out an extensive study on key tyre bales geometrical, mechanical, and hydraulic properties, revealing the current needs for further research: high-quality mechanical tests, including stiffness, shear strength, and creep. 

The above short literature review indicates that a very limited number of laboratory tests have been conducted to determine the mechanical properties of tyre bales for design purposes. Moreover, tests to date have been conducted on bales produced in the USA. However, tyres in the USA are generally larger than those used in Europe. Therefore further tests are required to enable a wider use of tyre bales in geotechnical applications. The lack of test data to define the basic mechanical properties of tyre bales led to prepare and implement the domestic laboratory-testing program to determine the basic mechanical properties required for the design of geotechnical structures with tyre bales. 

The laboratory tests performed at the Rzeszow University of Technology (RUT), included measurement and evaluation of full-scale tyre bales to determine the basic values for the geometry and unit weight, compressibility characteristics of tyre bales, including Young’s modulus and Poisson ratio, shear strength along the tyre–tyre and tyre–soil surfaces, creep and stiffness degradation under cyclic load. The respective test procedures and results of these tests are presented in the paper. The availability of data from such tests should encourage and accelerate the implementation of tyre bales in domestic geotechnical applications.

## 2. Materials 

### 2.1. Manufacturing of Tyre Bales 

Tyre bales used in the research program comprised whole waste passenger-vehicle tyres of R14-R17 type (diameter) compressed into a lightweight block. Each bale contained only one type and size of tyres, placed in the herringbone arrangement. The waste tyres used for bales production had no significant damages. Before bailing the tyres were cleaned. The fabrication of tyre bales was performed according to specification [23], which addresses the production, handling, storage, transport, and placement of standardized tyre bales. Apart from tyre bale dimensions—slightly modified to fit bailing machine capacity and to facilitate transport—the remaining production requirements, i.e., bale compressing force, mass/volume ratio, and number and tension of steel tie wires, were fulfilled.

Tyre bales were fabricated by a lightweight tyre-baling machine, designed and manufactured by the domestic producer. This baling machine typically compresses approximately 135 ± 1 waste tyres into a 2.0 m^3^ bale having a roughly estimated size of 2.0 ± 0.05 m × 1.3 ± 0.05 mm × 0.7 ± 0.05 m and weigh approximately 1030 ± 20 kg. Tyre baling results in a 4:1 volume reduction of loose tyres. The consistency of tyre bales production was ensured and indirectly controlled by the measurement of steel tie wires’ tension. Six 4 mm galvanized steel wires were spaced at 300 ± 20 mm along the length of the bale. Figure 1 shows a typical tyre bale used in the current research.

### 2.2. Tyre Bales Characteristics 

The dimensions were determined by taking five measurements each of the length *L*, width *B,* and height *H* of the bales and calculating the average of each dimension (Figure 2). The average volume *V* was determined by taking average values of the length, width, and height of the bales and multiplying them together. The weight *G* of a tyre bale was measured by a load cell system. The unit weight *γ_avg_* of the tyre bale is defined as the average volume measured. Table 1 provides the average dimensions, volume, weights, and unit weight measured and calculated for each tyre bale used in the current research.

### 2.3. Filling Material 

The uniformly graded medium size sand was used to simulate filling material applied in actual geotechnical applications. The medium sand was clean and the sieve analysis result according to standard [24] is presented in Figure 3. 

The direct shear test was conducted with the AB-2a direct shear apparatus according to standard [25]. The basic material characteristics of the medium sand used in this research program are as follows: Granulation (particle size): 0.25–1.00 mm;Coefficient of uniformity: 1.3;Coefficient of curvature: about 1.1;Void ratio: 0.59–0.83;Moisture: 12.6–12.8%;Specific gravity: 2.66 g/cm^3^;Friction angle: 35.2°;Bulk weight (density): 19.0 ± 0.5 kN/m^3^.

## 3. Methods 

### 3.1. Compressibility Test 

A series of compression tests were conducted to evaluate the behavior of the tyre bales when subjected to vertical normal loads and to determine Young’s modulus and Poisson’s ratio for tyre bales. Compression tests on the tyre bales were conducted following the standard [26]. All six specimens (full-size tyre bales) P-1 to P-6 were used in the compression test. The top and bottom loading platens consisted of three concrete slabs (two bottom and one top) with dimensions 0.15 m × 1.50 m × 3.00 m, that extended in all directions beyond the edge of the fully loaded specimen (Figure 4). Additional 65-mm thick steel plate and steel bearing with the diameter of 200 mm were used to ensure appropriate loading transfer.

The computer-controlled hydraulic actuator with a capacity of 630 kN was used to apply the vertical load with a speed of 0.5 kN/s. The load scheme uses three load levels up to a force of 50, 100, 250 kN and six cycles of load-unload tests before applying the failure load. The example of the load track for P2 specimen is showed in Figure 5. The vertical deformation measurements were made utilizing four LVDT transducers with full stroke range up to 300 mm, located in four corners of the top platen within equal distances around the specimens (Figure 6). Vertical deformation was also measured in the center of a concrete slab/steel plate set-up using the actuator’s piston displacement. The horizontal deformations were measured on one tyre bale’s side using three LVDT transducers with full stroke range up to 100 mm. To remove errors due to curvature of the tyre bales small steel pads were mounted to the respective bale’s side. Load and displacement data were recorded automatically during testing at 2.0 Hz frequency using the HBM Spider data acquisition system.

### 3.2. Full-Scale Direct Shear Tests 

A full-scale direct shear testing was carried out to determine the interface strength between bales (tyre–tyre interface test) and between bales and soil (tyre–soil interface test). These tests are similar to that of a soil direct shear test, in which the material is loaded to fail horizontally along a defined shear plane according to standard [25]. The tyre bale interface is defined as the tyre material around the perimeter of the bale, which can be characterized as an irregular and variable surface. 

#### 3.2.1. Tyre–Tyre Interface Test 

The tyre–tyre interface test procedure involved stacking three bales and applying a normal and shear load to one bale while holding the others stationary (Figure 7). The top (sliding) bale was shifted horizontally with a hydraulic actuator. The horizontal shear load was applied at a distance of 0.25 m from the bottom of the sliding bale (1/3 the height of the bale), which represented the resultant force due to a triangular stress distribution on the tyre bale. Four sliding bales P-1 to P-4 were applied in the tyre–tyre interface test, while bales P-5 and P-6 were held stationary and used as the bottom bales. A steel channel C-300 was placed on the loaded face of the sliding bale to distribute the load applied by the actuator. The traditional direction of loading, i.e., shear in the direction parallel to the bottom baling wires, was assumed because it would provide the highest shear resistance [22].

The computer-controlled hydraulic actuator with a medium stroke (400 mm) and capacity (44 kN) was used to apply the horizontal load. The normal load was applied gradually by placing additional weight on the sliding bale to increase the normal load. The following stages of normal loading *V* were applied subsequently:
(a)Stage I–the dead weight of the sliding bale (*V_I_* = 10.0 kN);(b)Stage II–the additional weight of steel plate placed on the sliding bale (*V_II_* = 13.4 kN);(c)Stage III–the additional weight of a tyre bale placed on the sliding bale (steel plate was removed) (*V_III_* = 20.0 kN); (d)Stage IV–the additional weight of concrete road slab placed on the additional tyre bale (*V_IV_* = 26.2 kN). 

The failure shear load is typically determined using two different methods: the bi-linear method and the maximum compression method. For the bi-linear method, the load-displacement curves are approximated as bi-linear curves to determine the peak shear load [22]. Shear failure is defined as the intersection between the two lines that can be approximated for each bi-linear portion of the curve. However, La Rocque [19] defined shear failure as the slipping load at the relevant bale compression. The latter method was used in this study with modification, which enabled the independence of results from surface irregularities. 

The horizontal load (H) was applied with a constant speed of 4 mm/min until slippage occurs, i.e., the substantial horizontal displacement of sliding bale was observed with a noticeable decrease of the horizontal load or the horizontal displacement exceeds 8–10% of the bale length, i.e., 160 ÷ 200 mm. After a slippage had been observed, the horizontal loading was stopped and the actuator piston’s displacement was maintained about 5 min until the load stabilized. This stabilized value of the horizontal load was further considered as the failure shear load. Subsequently, the specimen was unloaded with a constant speed of 10 mm/min. Finally, the normal load was removed and changed according to the testing program, and the next stage of testing was started. In fact, the failure shear load depends on a loading speed. In a real geotechnical structure, the shear-initiating speed is very small within the range of 0.001 mm/min to 0.1 mm/min. Therefore the horizontal shear load measured at the load stabilization can be assumed as the minimum failure load-inducing slippage. Thus the real behavior of a tyre bale built-in an actual geotechnical structure is simulated.

Six LVDT transducers with full stroke range up to 300 mm were placed at the front (4 sensors) and rear (2 sensors) of the sliding bale to measure the horizontal displacements. The load cell combined with the actuator was used to measure the horizontal load applied to the sliding bale. Load and displacement data were recorded automatically during testing at 0.5 Hz frequency using the HBM Spider data acquisition system.

#### 3.2.2. Tyre–Soil Interface Test 

After the tyre–tyre interface test had been carried out, the test set-up was modified so that a soil layer could be compacted beneath the sliding bale to determine the tyre bale-soil interface strength (tyre–soil interface test). The modified procedure involved stacking two bales; the bottom bale was held stationary in the self-supporting wooden box with dimensions of 2.4 m × 1.8 m × 1.0 m and filled with soil (Figure 8). The thickness of the soil layer was chosen as about 130 mm to simulate the typical tyre-soil structure composition and to ensure that failure occurs along the interface and not within the soil. The medium size sand as filling soil was chosen for the testing program (see p. 2.3). Compaction of the soil layer in the box was accomplished with the Wacker Neuson BS-600 rammer with the foot size 280 mm × 336 mm and the total mass of 66 kg. To simulate wet conditions within actual tyre–soil structure the tyre bale interface and soli layer were wetted after placing in water by a sprinkler. 

The sliding bale was placed on the soil layer and shifted horizontally with a hydraulic actuator. In this case, the horizontal shear load was applied at a distance of about 0.38 m (1/2 the height of the bale) from the bottom of the sliding bale. An L-shaped 20-mm thick steel plate was placed on the sliding bale; the vertical part of the plate distributed the load applied by the actuator on the loaded face of the sliding bale. 

Three sliding tyre bales (specimens) P-1, P-2, and P3 were applied in the tyre–soil interface test, while bale P-6 used as the bottom bale was held stationary in the box. The applied shear load *H* was the same as in the previous tyre–tyre interface test. However, the normal load *V* stages were slightly changed as follows:
(a)Stage I–the dead weight of the sliding bale and the steel plate (*V_I_* = 22.2 kN);(b)Stage II–the additional weight of a concrete road slab placed on a steel plate (*V_II_* = 28.4 kN); (c)Stage III–the additional weight of a concrete road slab and a tyre bale placed on a steel plate (*V_III_* = 38.4 kN);(d)Stage IV–the additional weight of two concrete road slabs placed on the steel plate (*V_IV_* = 44.0 kN).

The same criteria and methodology as previously (i.e., in the tyre–tyre tests) were used for the determination of the failure shear load. Each specimen was tested three times. Eight LVDT transducers were placed at the front (2 sensors), rear (2 sensors), and sides (4 sensors) of the sliding bale to measure the horizontal and vertical displacements (Figure 8). Load and displacement data were recorded automatically by testing similarly as previously.

### 3.3. Creep Test 

The tyre bale creep tests consisted of placing a constant load on a single tyre bale and monitoring the time-dependent deformations under sustained load for a period of up to 165 h at a constant room temperature. According to La Rocque [19] and Zornberg et al. [21] creep curves are approximately linear in semi-log space for testing times up to 1000 h, indicating that creep tests did not need to be performed for more than a few days to properly characterize the behavior. The test setup is shown in Figure 9. The tyre bale was placed on a concrete slab with nominal dimensions 0.15 m × 1.50 m × 3.00 m, which provided the reaction, while subsequent four concrete slabs were weighted and used as a loading. Single slab weighted from 15.6 kN to 16.5 kN and the total value of sustained load was 64.6 kN. 

Three tyre bales P-1 to P-3 were applied in the creep test. Each tyre bale was tested with a gradually applied load up to 64.6 kN, equivalent to vertical stress of 23.6 kPa (based on the cross-sectional area of the bale) for a period of about 3, 4, and 7 days, respectively. This stress was determined for a 0.8 to 1.0 m of compacted soil cover (typical road sub-base) and a 0.3-m thick asphalt pavement layers over the bales. The load was applied gradually slab-by-slab with intervals in-between to enable stabilization of bale deformation under the respective load. The stabilization was reached when the difference of bale deformation in 1 h period was less than 0.03 mm. The average stabilization time was about 24 h. 

The vertical deformation measurements were made using four LVDT transducers with full stroke range up to 100 mm. The sensors were located in four corners of the bottom loading platen within equal distances around the specimens (Figure 9). Displacement data were recorded automatically during testing at 1.0 Hz frequency using the HBM Spider data acquisition system.

### 3.4. Cyclic Load Test 

Before performing the compression tests, cyclic load tests were performed. These tests were carried out to evaluate deflection under traffic and to provide a pseudo evaluation of resilient modulus. Three specimens P-1 to P-3 were applied in the cyclic test and placed on two bottom concrete slabs (Figure 10a). A rigid circular 30-mm thick steel plate of 300 mm in diameter was used to simulate a standard single wheel load. A level bearing surface was provided beneath the circular plate using a non-shrink mortar mix (Figure 10b). 

The maximum load of 20 kN was applied at a 0.4 and 0.8 Hz frequency to simulate traffic loading (i.e., a standard single wheel load over a typical pavement structure about 800-mm thick). The test was performed for 10 and 16 thousands of load cycles (Table 2). Load and vertical displacement data were measured directly by the actuator and were recorded automatically during testing at 10.0 Hz frequency using the HBM Spider data acquisition system.

## 4. Results and Discussion

### 4.1. Compressibility Test

The load versus vertical and horizontal deformation plots are given to characterize the bales behavior under compression. In Figure 11a,b the typical behavior of the exemplary P-4 bale is presented for vertical and horizontal deformations, respectively (the sixth cycle, *p* = 250 kN). The considerable differences between displacement records for individual LVDTs are shown, particularly for horizontal ones. It is not due to uniform loading of a bale despite stiff platen applied as well as different structural homogeneity of the bales themselves. The summary average displacement plots for all bales are given in Figure 12a,b, for vertical and horizontal deformations, respectively (the sixth cycle, *p* = 250 kN). 

The considerable differences in bale’s behavior under compression for individual bales are revealed. This was attributed to the variability of the bales’ structure and the fact that no two tests, although using the same type of bales, were the same. For all bales, the non-linear response of the tyre bale was revealed as well as a hysteretic unloading curve was obtained. The hysteretic curvature implies that the deformation remains in the tyre bale structure even after the applied load is removed, although the sixth load cycle was applied. This is evident by the apparent plastic deformation of the tyre bale after unloading, in which approximately 15 mm of deformation remain, most of which is fully recovered to a value of 2–3 mm after 5 min. The hysteretic curvature with approximately ten times smaller apparent plastic deformations was also observed in the horizontal direction.

The excessive platen incline was treated as a failure in each scheme (Figure 13a). The reliable record of vertical displacements was impossible due to platen rotation and the applied force was recognized as a maximum (failure) load *P_max_*. After reaching *P_max_* the tyre bale was unloaded. Although no bale’s steel tie wires broke at failure load, the decrease of wire tension was observed (loose wires) after unloading (Figure 13b). 

The failure loads and average vertical and horizontal displacements at *p* = 250 kN and at failure load, based on records of four vertical and three horizontal LVDTs, respectively, are collected together in Table 3. The compressive strength of the tyre bales used in the current research was in the range of 355–436 kN and the average strength obtained was 382 kN. At this failure load, the average vertical displacement was 125 mm, which means 16.7% average vertical bale’s deformation. However, after unloading the bales had a permanent vertical deformation in the range of 25–30 mm only, which revealed their good elasticity under compression and the possibility of almost full recovering in several min after unloading.

The stress–strain curves for all bales obtained in the final (failure) cycle of loading are shown in Figure 10. The relevant values of stresses and strains were calculated as follows:(1)$$\sigma_{v}=\frac{P_{max}}{A},$$
and
(2)$$\varepsilon_{v}=\frac{v_{avg}}{H},$$
where:*P_max_*–failure load (Table 3);*v_avg_*–average vertical displacement at failure load (Table 3);*A* and *H*–area and height of the bale (Table 1).

The comparison of the five curves in Figure 14 provides evidence of the effects of tyre bale structure and variability between bales. The test results indicated that the tyre bales did not have the ultimate strength at stress levels of less than 130 kPa, while the maximum ultimate strength obtained was 158 kPa. The higher strength was unable to obtain due to excessive platen inclination under ultimate loading. The vertical strain of the tyre bales was over 30% at the maximum test loads. However, at working stress levels (typically less than 50 kPa) strains are much lower and will be further reduced by filling and bale confinement. The ultimate strength of tyre bales can be enhanced due to sand infill (reduction of empty spaces in bales) and lateral constrains of a single bale, located always in a group of bales. The tests, which simulated both aforementioned conditions, revealed that the ultimate strength of tyre bales could be enhanced up to 815 kPa [21].

The Young’s modulus (Ev) and the Poison’s ratio (ν) for a tyre bale are the basic design parameters of high importance, required for further site implementation. Therefore the evaluation of both parameters has been carried out based on compressibility test results. The non-linear deformation of the tyre bale can be represented by either an average linear modulus over the stress range of interest or a series of secant moduli with the applied load. The first approach was applied in the current research to assess the approximate modulus value and the latter was applied to show modulus variability under increasing load. 

The Young’s modulus for each stress range considered was assessed according to the following formula:(3)$$E_{V}=\frac{\sigma_{V}}{\Delta \varepsilon_{v}},$$
where:$\sigma_{v}$—maximum stress in the range considered;$\Delta \varepsilon_{v}$—vertical strain increase in the range considered.

The values of the average Young’s moduli for the deformation curves of the tyre bales are presented in Table 4. The approximate Young’s modulus for the tyre bale is in the range of 713.5 kPa to 904 kPa at a medium stress level of about 90 kPa (*p* = 250 kN). However, the average *E_v_* value 826 kPa is close to the values reported in [19,21,22] for a similar test setup (i.e., 914 kPa, 831 kPa, and 766 kPa, respectively). Moreover, including a predicted value of the tyre bale stiffness with sand infill, the determined modulus could be increased by 30% according to [21]. To show Young’s modulus variability under increasing load a series of secant moduli with applied load was determined and graphically presented for P-5 bale in Figure 15.

The Poisson’s ratio was defined as the horizontal strain divided by the total vertical strain of the tyre bale’s material. The equation for the Poisson’s ratio determination is as follows:(4)$$\nu =\frac{\varepsilon_{h}}{\varepsilon_{v}}\;,$$
where:$\varepsilon_{h}$—maximum horizontal strain in the range of 0–250 kN;$\varepsilon_{v}$—vertical strain in the range of 0–250 kN. 

The maximum horizontal strain *ε_h_* was calculated on the assumption of bales symmetry as the double horizontal deformation *2h_avg_* obtained at *P_max_* (Table 3), divided by the average bale length *L* (Table 1): (5)$$\varepsilon_{h}=\frac{2h_{avg}}{L},$$
where:$h_{avg}$—horizontal deformation (Table 3);L—average bale length (Table 1). 

It should be noted that horizontal strain was calculated only for the bale’s deformation in the length direction (*L*), because the perpendicular deformation (*B*-direction) was restrained from the wire ties. The horizontal deformation measurements indicated a relatively low Poison’s ratio on the order of 0.11 at the medium stress level of about 90 kPa (*p* = 250 kN). Lateral movement at low and medium stress levels is likely restricted by the combination of compression during baling and vertical orientation of the tyres in the bale. Values reported in [21] ranged from 0.1 to 0.2 for stresses less than 47.9 kPa, and increased up to 0.3–0.4 at higher stresses and those reported in [22] ranged from 0.08 to 0.24 at maximum stress of 31.1 kPa. The Poisson’s ratio variability under increasing load in the range of 0–250 kN is graphically presented for P-5 bale in Figure 16. 

It should be noted that the asymptotic value of Poisson’s ratio was not determined yet, thus the values given in Table 4 may be slightly underestimated. Finally, it should be underlined once again, that tyre bales used in the actual field applications will be surrounded by other tyre bales or soil, and therefore the effect of the confinement on the compressibility of the tyre bale needs to be taken into account.

### 4.2. Full-Scale Direct Shear Tests 

#### 4.2.1. Tyre–Tyre Interface Test

A series of horizontal load-displacement curves was plotted for each of the specimens to observe the development of shear resistance along the interface as well as to determine the failure shear load along the interface. The typical curves for P-4 specimen under the normal load in stage I (*V_I_* = 10 kN) are shown in the following figures. 

In Figure 17 three distinct sections can be observed along the load–time curve: (I) loading up to slippage as defined above, (II) load stabilization, and (III) unloading. The average (of two sensors) front displacement of the bale was used to plot the *H–*Δ curve since it was maintained at a constant rate throughout the test and because the rear displacement also included compressions of the bale. The local load drops are observed in the *H–*Δ curve due to irregularity of tyre bales’ surfaces. 

Shear stress versus bale displacement curves for the subsequent normal loads (load stages) are shown in Figure 18. Regardless of the applied normal (vertical) load, all direct shear tests conducted in this study showed a similar trend. The shear *τ* and normal *σ* stresses along the tyre bale interface were defined as the applied load divided by the average area of the tyre bale (Table 5). 

The failure shear load observed in the test results was used to define the shear strength envelope, that corresponds to the failure points of 12 tests conducted in this study. The shear strength envelope is shown in Figure 19 as a function of the applied normal stress. A linear trend line was used to represent a failure envelope passing through the points. To derive the shear strength of a tyre–tyre interface, based on the direct shear test according to standard [25], the Mohr-Coulomb criterion (6) is typically used: (6)τ=σ⋅tgϕ+c
where:*τ*—shear strength;*σ*—normal stress;*ϕ*—friction angle of the material (in this case: friction at the tyre–tyre interface);*c*—cohesion of the material (typically, sandy soils are considered cohesion-less). 

The interface shear strength parameters (*c* and *ϕ*) were determined from linear regression of the test results as *ϕ* = 46.0° and *c* = 0.03 kPa. The curve fitting is quite good as the coefficient of determination R^2^ equals 0.944. The approximate shear stresses listed in Table 5 were obtained using this function. Error estimation, calculated as a deviation of the actual test result from the envelope function, is within the range of 0.11% to 21.53% (Table 5). The variability of these test results was not more than 25%, but only five results had a deviation of more than 10%. Thus the variability in the data set was moderate, indicating an acceptable variability of shear strength from bale to bale. It may be concluded that the interface shear strength of tyre bales is comparatively high. However, the effect of moisture at the interface and the effect of the directionality of tyre bale placement were not evaluated as part of this study. 

#### 4.2.2. Tyre–Soil Interface Test 

As previously, a series of horizontal load–displacement curves were plotted for each of the specimens to evaluate the development of shear resistance along the tyre–soil interface as well as to determine the failure shear load along the interface. The typical curves for P-2 specimen under the normal load in stage IV (maximum normal load *V_IV_* = 44.0 kN) are shown in Figure 20. The load—time behavior of the soil interface is similar to that of the tyre bale only interface, however, no local load drops are observed in the *H–*Δ curve. In these tests, the irregularity of the stationary tyre bale’s surface was levelled by the soil cover. 

Typical interface shear stress versus displacement curves for P-2 specimen in all load stages are shown in Figure 21. The stress–displacement behavior of the soil interface is similar to that of the tyre bale only interface; regardless of the applied normal (vertical) load, all direct shear tests conducted in this study showed a similar trend. All curves are also smoother than in case of tyre–tyre tests; no drops are observed due to regular sand interface. However, for the medium sand layer, there is an increase of the displacement of the sliding bale before reaching the failure strength when compared with the tyre bale only interface (increased from 1.5–2.5% of the bale length for the tyre bale only interface up to 4.0–6.0% for the tyre-soil interfaces).

The shear *τ* and normal *σ* stresses along the tyre bale interface were defined as the applied load divided by the average area of the tyre bale (Table 6).

The shear strength envelope is shown in Figure 22 as a function of the applied normal stress. The interface shear strength parameters determined from linear regression of the test results are as follows: *ϕ* = 29.6° and *c* = 0.77 kPa. The curve fitting is even better than for the tyre–tyre interface since the coefficient of determination R^2^ equals 0.962. Error estimation is within the range of 0.18% to 11.8% (Table 6). The variability of these test results was not more than 12%, but only three results had deviation of more than 5%. Thus the variability in the data set was very small. Similarly to the tyre bale only interface, the shear strength of tyre–soil interface is quite high. This time the effect of moisture at the interface was included in the evaluation.

For the tyre bale–sand interface used in this study, the friction angle *ϕ* is about 15% lower than the friction angle measured for the medium sand in the direct shear test. The comparison of the tyre bale–sand interface strength and sand only strength indicates that the frictional response along the tyre bale interface cannot be directly predicted by the direct shear testing of the soil. The decrease in strength is due to the interface strength defined along the footprint area, which is significantly larger than the actual area. The 35% lower friction angle of the sand fill, when compared to the tyre only interface, is due to the reduced contact area between the tyre ridges caused by the sand infill filling the voids. Moreover, the effect of moisture at the interface was included in the evaluation on the contrary to tyre–tyre tests.

### 4.3. Creep Test 

The full creep curves for all tested specimens are shown in Figure 23. In Table 7 the deformation and time data are collected along with the regression analysis results (fitting curves and coefficient of determination R^2^) estimated for all particular periods of creep loading. The following symbols were used in Table 7: *h_0_*—initial bale depth, *t*—load time in h, Δ*h_exp_*—the experimental measurement of creep deformation (an average of four records), R^2^—coefficients of determination, Δ*h_365_*—estimated 1-year creep deformation, *c_α,365_*—estimated 1-year creep coefficient. The Δ*h_365_* values were estimated using the fitted regression lines given in Table 7. The regression lines are very well fitted to testing results with the coefficients of determination R^2^ > 0.94, except one case for specimen P-3 (load slab no.1), when test experienced accelerated movement on one corner after loading. Progressive deformation in different regions of the P-3 specimen resulted in variation in both deformation and load and made the determination of creep strain rate difficult.

The average creep deformation of three specimens under four-phase sustained load for the average sustain load period of 110 h was about 40 mm, which constitutes 6.1% of the average height of the bale. Approximately 95% of the deformation occurred on the first day and the maximum deformation appeared to occur within three days. The 1-year creep coefficients *c_α,_*_365_ were estimated as 0.0048, 0.0042, and 0.0035 for P-1 to P-3 specimens, respectively, and the average *c_α,_*_365_ value for the creep test as 0.0039 was obtained. This value is similar to those obtained by La Rocque [19] and Zornberg et al. [21]; the difference is less than 5%. The test indicates relatively little creep response at long-term stress levels. The post-3-day movement would roughly be the anticipated post-construction movement and could be reduced by preloading.

### 4.4. Cyclic Load Test

The cyclic load test results for three tyre bales in the form of time–cyclic displacements plots are shown in Figure 24.

The minimum deformation represents permanent deformation in the tyre bale and appears to be fairly similar for P-1 and P-2 bales in the same load conditions. The maximum deformation minus the minimum deformation, representing a resilient value, was about 22 mm and 10 mm for P-1 and P-2 bale, respectively. However, only the latter value can be treated as a true one considering a resilient value because in the P-1 case a mortar layer beneath the circular plate was accidentally destroyed due to overloading. This resilient value was almost constant in the entire period of test time, i.e., about 6.6 h (400 min), revealing no stiffness degradation of tyre bales under cyclic loading. The temporary peak in the plot for P-1 bale is due to the sudden and short-term decrease of ambient test temperature (about 1.5 °C), which induced the increase of bale’s stiffens. In contrary to previous cyclic tests, the resilient value for P-3 bale was about only 1 mm due to four times smaller load amplitude compared to P-1 and P-2 cases. However, also in this case after about 1 h of loading the resilient value was almost constant in the entire period of test time, showing no bale degradation.

## 5. Summary

The results of the testing program presented in the paper illustrate the mechanical properties of tyre bales required for geotechnical projects. A brief summary of the obtained results is provided below along with the qualitative comparison with the relevant values found in the literature [7,8,19,20,21,22].


The cuboid tyre bales comprising approximately 135 car tyres are used in the research. The baling machine produced tyre bales of approximate dimensions 2.05 m × 1.30 m × 0.75 m, a mass of around 1030 kg, the average volume of around 2.0 m^3^, and the unit weight defined using the average volume of approximately 0.515 Mg/m^3^. This unit weight was found to be slightly lower than the respective values reported in the literature.The approximate Young’s modulus for the tyre bales determined on the basis of the compression testing is 826 kPa and is close to the values reported in the literature. The horizontal deformation measurements indicate a relatively low Poison’s ratio on the order of 0.11, while these values reported in literature ranged from 0.08 to 0.24.The results from the dry tyre–tyre interface testing can be combined and modelled with a linear failure envelope with the cohesion of 0.03 kPa and friction angle of 46.0°, showing moderate variability between different bales within the range of 0.11% to 21.53%. The corresponding results from the wet tyre-soil interface testing indicate that for medium sand a linear failure envelope can be estimated with a small cohesion 0.77 kPa and a friction angle of 29.6° with the small variability of test results not more than 12%. Both results match well with the values provided in the literature.The tyre–soil interface strength determined in the test is about 15% weaker than the medium sand strength, while up to 20% reduction was revealed in the literature.Creep deformation due to sustained normal compressive load for up to five days of loading constitutes 6.1% of the average height of the bale. A significant portion of the creep deformation (approximately 95%) occurred in the first day and the maximum deformation appeared within three days. The 1-year creep coefficient was estimated as 0.0039 and this value is very similar to those obtained in the literature.The cyclic load test results obtained as a resilient value (the difference between maximum and minimum deformations) within about 400 min of loading revealed no stiffness degradation of tyre bales.


These results can be directly applied to design and construct tyre-baled structures. Based on the aforementioned testing results the first Polish application of abutment backfill from the tyre bales in a road bridge was designed and executed [18].

## Figures and Tables

**Figure 1 materials-14-03310-f001:**
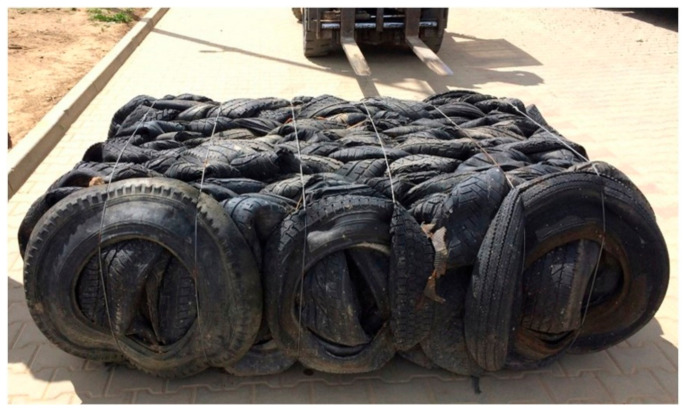
Typical tyre bale used in the current research.

**Figure 2 materials-14-03310-f002:**
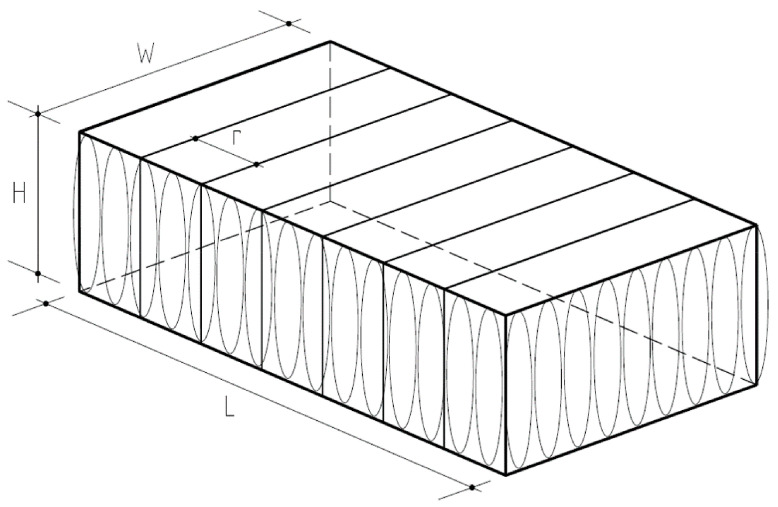
Scheme of basic dimensions of a tyre bale.

**Figure 3 materials-14-03310-f003:**
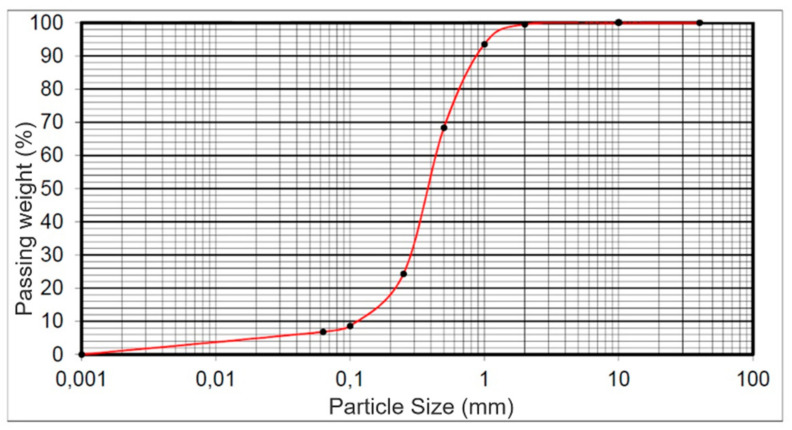
The sieve analysis result for medium sand according to [24].

**Figure 4 materials-14-03310-f004:**
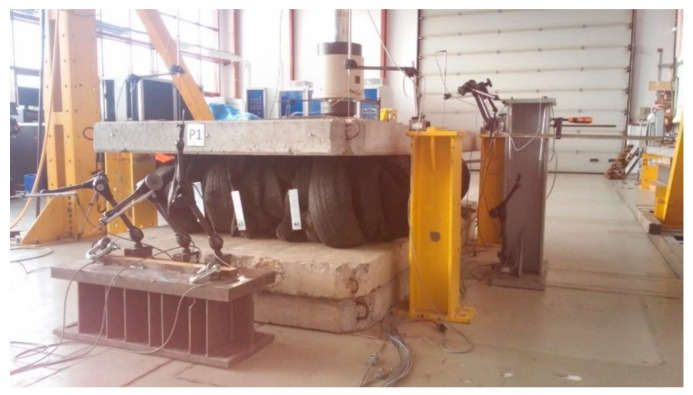
Compression test stand.

**Figure 5 materials-14-03310-f005:**
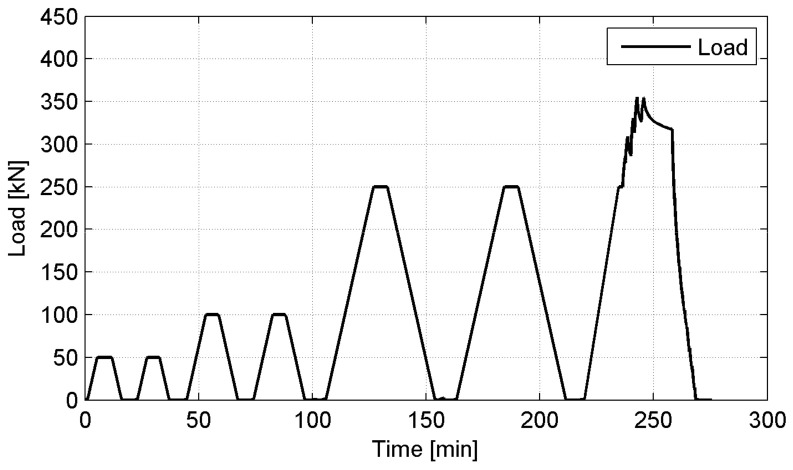
Typical load track for tyre bale compression.

**Figure 6 materials-14-03310-f006:**
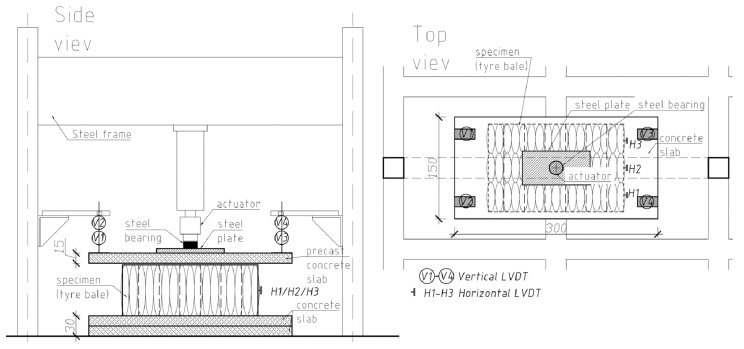
Compression test set-up.

**Figure 7 materials-14-03310-f007:**
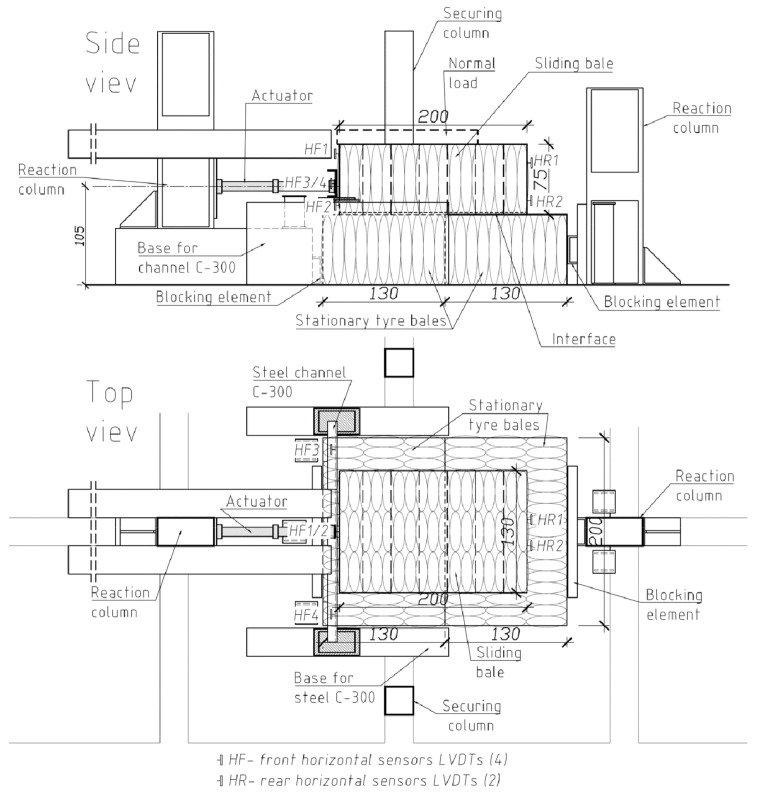
Scheme of tyre–tyre interface test (without additional normal loading).

**Figure 8 materials-14-03310-f008:**
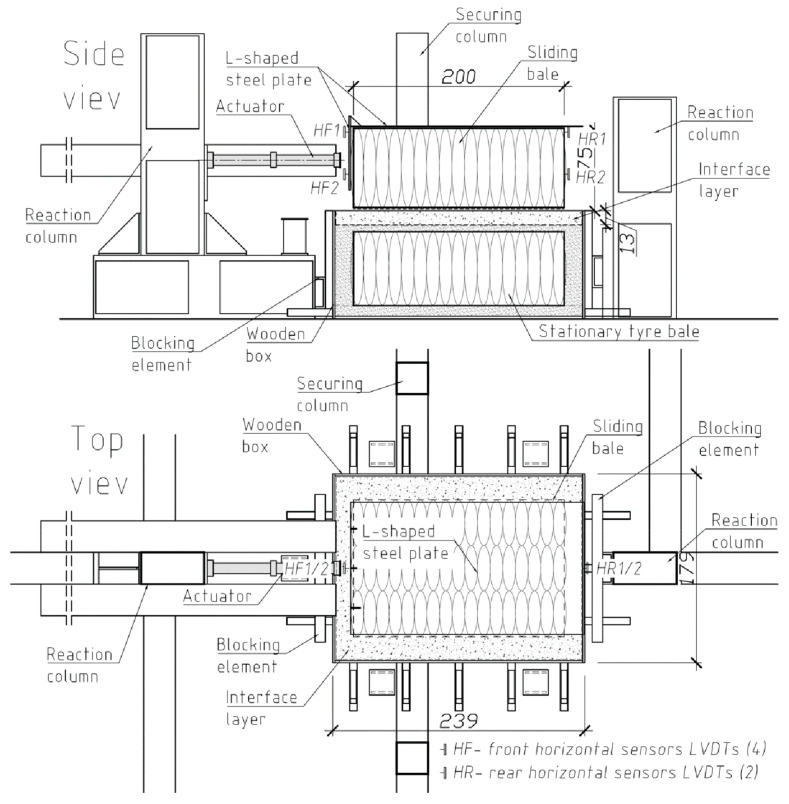
Tyre bale–soil interface test stet-up (without additional normal loading).

**Figure 9 materials-14-03310-f009:**
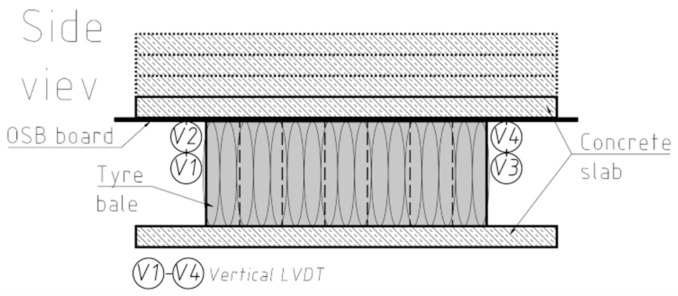
Creep test set-up.

**Figure 10 materials-14-03310-f010:**
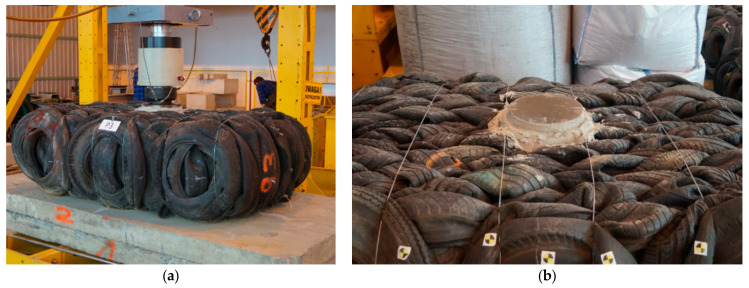
Cyclic test stand: (**a**) overall view; (**b**) bearing surface.

**Figure 11 materials-14-03310-f011:**
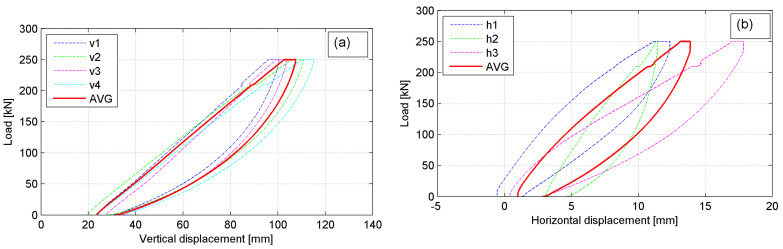
The behavior of P-4 bale under load *p* = 250 kN: (**a**) vertical displacements; (**b**) horizontal displacements.

**Figure 12 materials-14-03310-f012:**
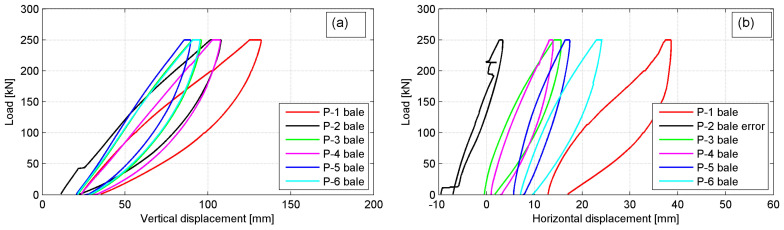
Average displacement plots for all bales under load *p* = 250 kN: (**a**) vertical displacements; (**b**) horizontal displacements.

**Figure 13 materials-14-03310-f013:**
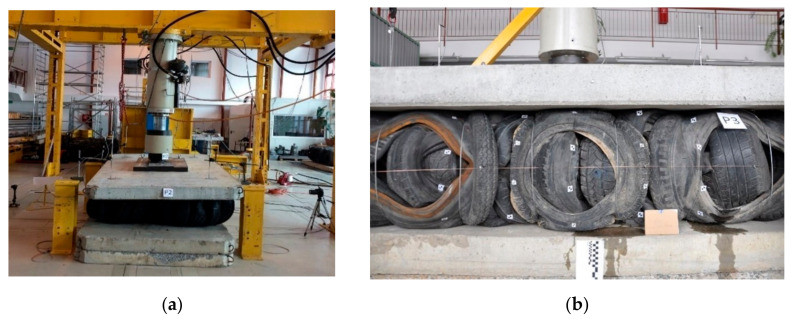
The typical failure mode of tyre bales under compression: (**a**) excessive platen rotation at failure load; (**b**) loose side wires after unloading.

**Figure 14 materials-14-03310-f014:**
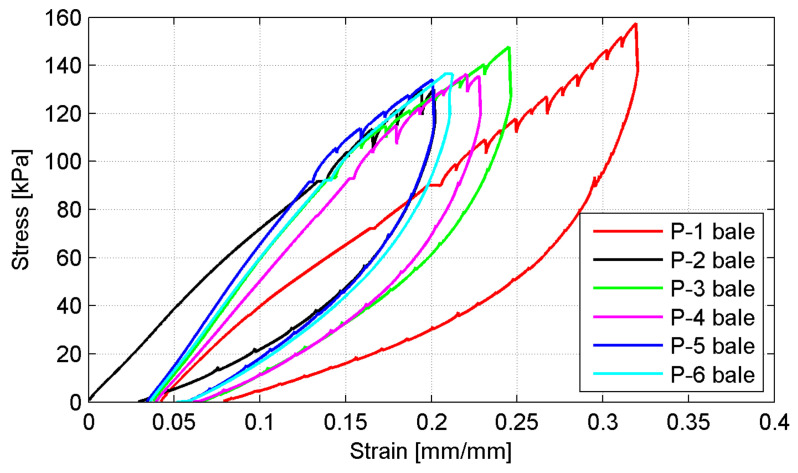
Stress–strain plots for all bales under failure load.

**Figure 15 materials-14-03310-f015:**
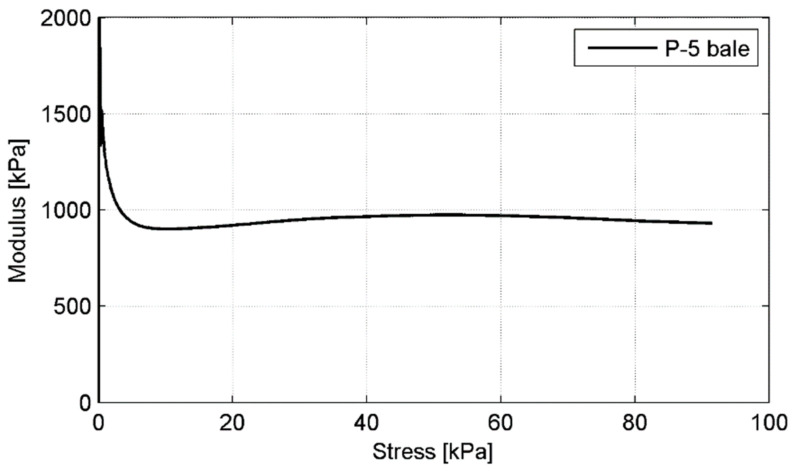
Young’s modulus variability for P-5 bale under increasing load.

**Figure 16 materials-14-03310-f016:**
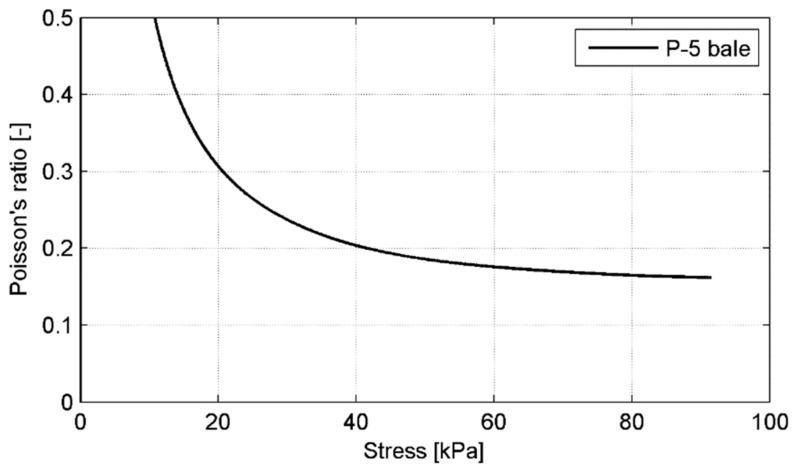
Poisson’s ratio variability for P-5 bale under increasing load.

**Figure 17 materials-14-03310-f017:**
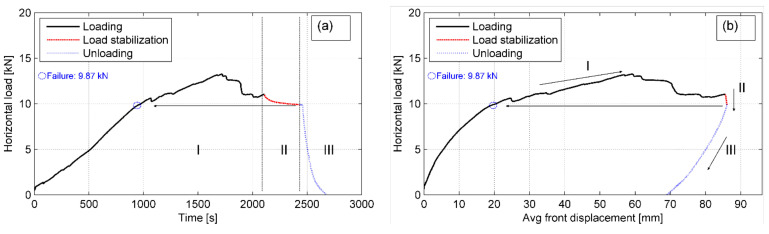
Typical curves for P-4 specimen in stage I: (**a**) load–time curve; (**b**) load–displacement curve for average front displacement.

**Figure 18 materials-14-03310-f018:**
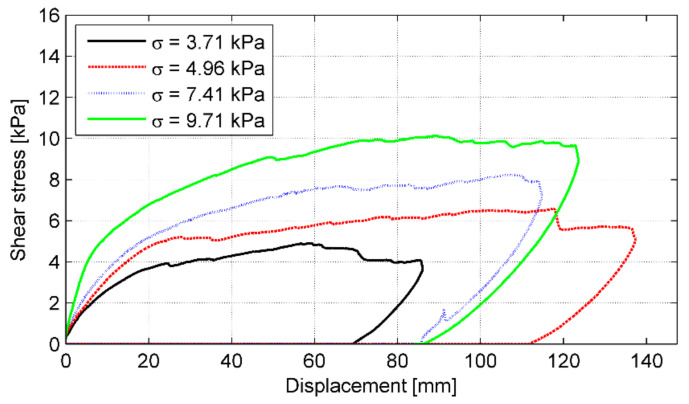
Typical shear stress–displacement curves in subsequent load stages for P-4 specimen.

**Figure 19 materials-14-03310-f019:**
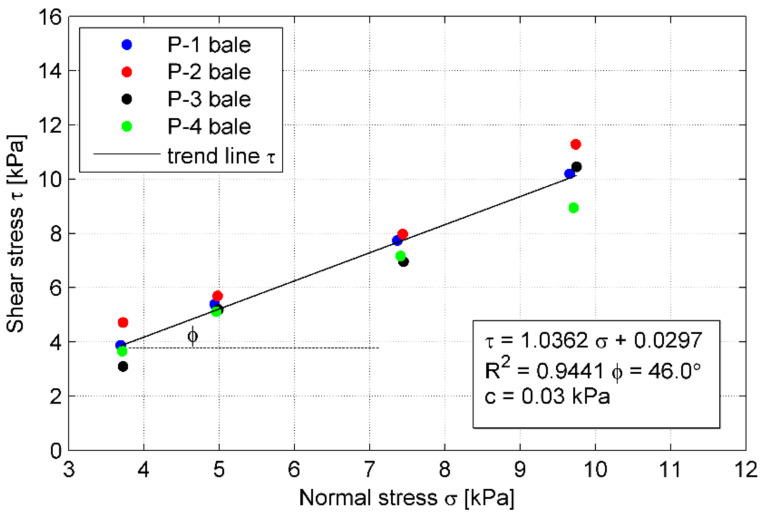
Shear strength envelope for tyre–tyre interface.

**Figure 20 materials-14-03310-f020:**
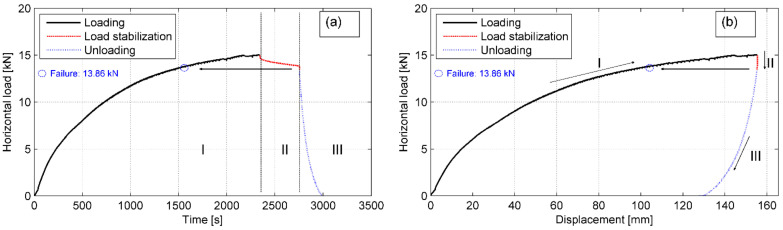
Typical curves for P-2 specimen in stage IV: (**a**) load–time curve; (**b**) load-displacement curve for average front displacement.

**Figure 21 materials-14-03310-f021:**
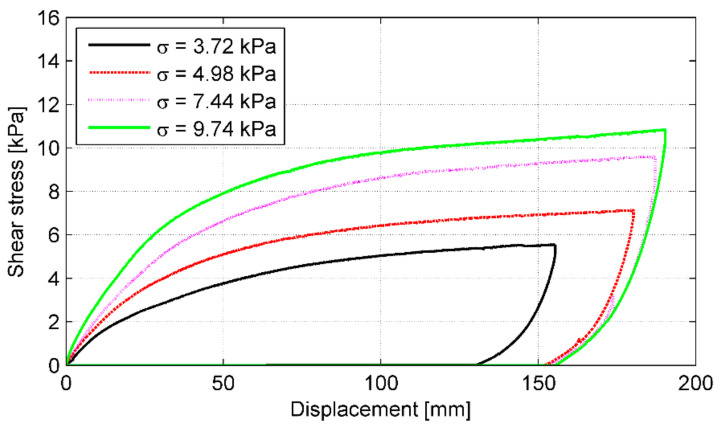
Shear stress–displacement curves for average front displacements in subsequent load stages for P-2 specimen.

**Figure 22 materials-14-03310-f022:**
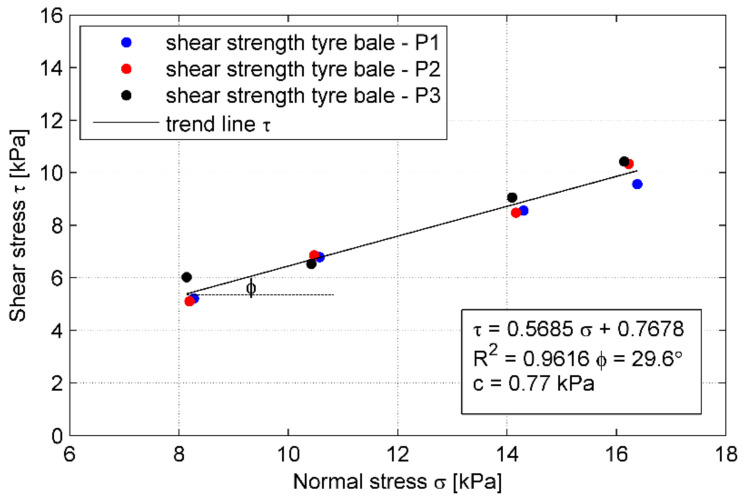
Shear strength envelope for tyre–soil interface.

**Figure 23 materials-14-03310-f023:**
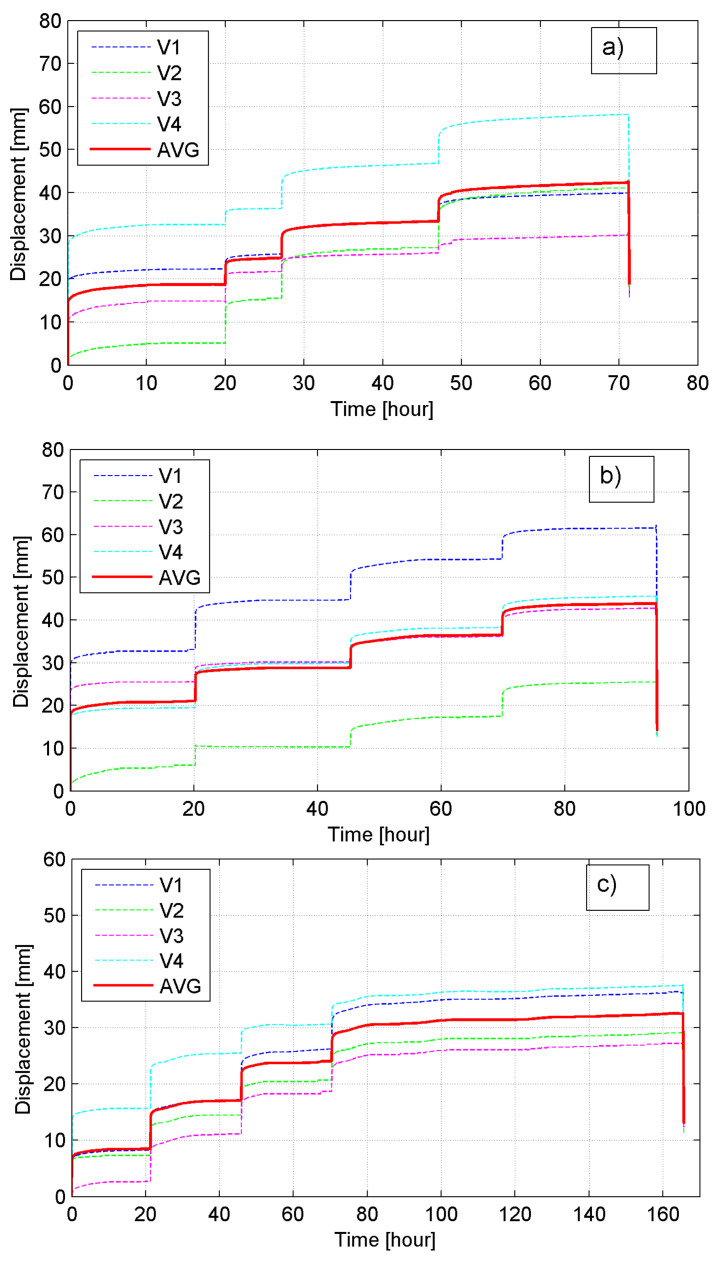
Creep curves for: (**a**) P-1 specimen; (**b**) P-2 specimen; (**c**) P-3 specimen.

**Figure 24 materials-14-03310-f024:**
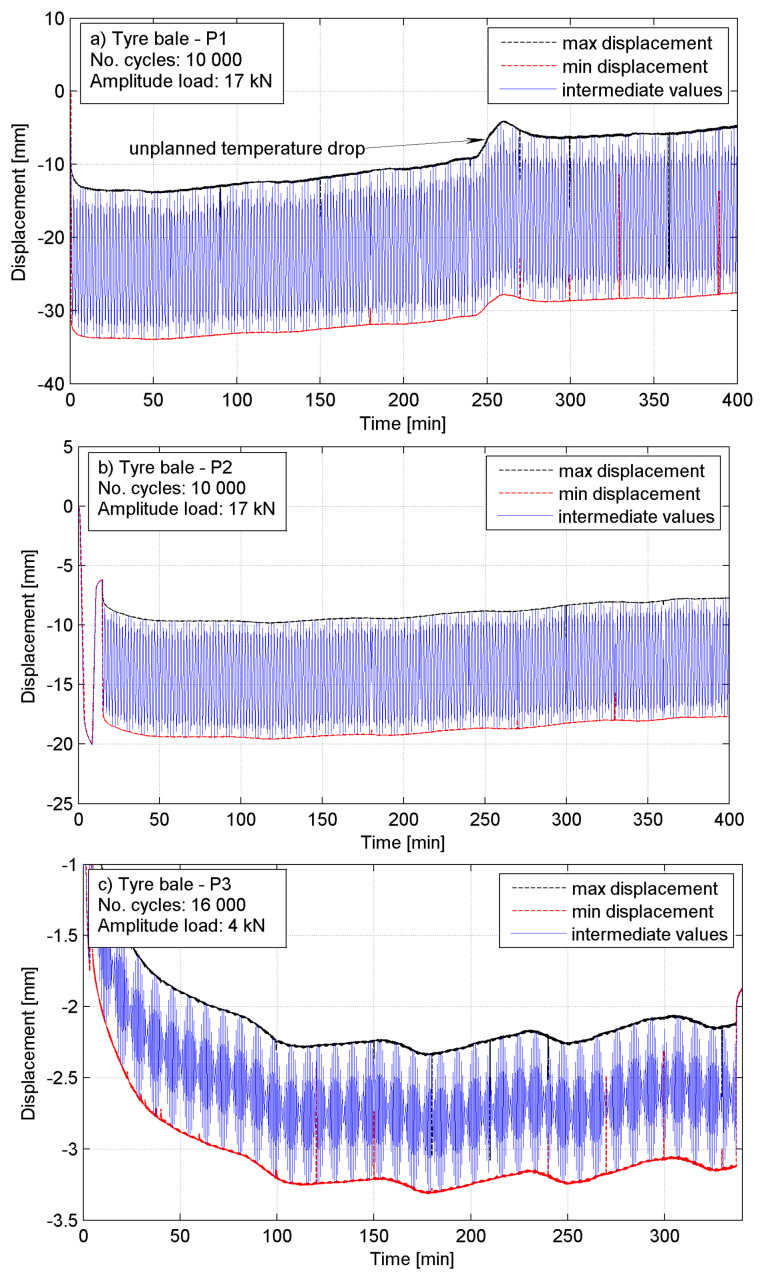
Time–displacements plots for subsequent testing bales.

**Table 1 materials-14-03310-t001:** Basic dimensions and weights of tyre bales used in the current research.

Bale No.	Number of Tyres	Length *L*	Width *B*	Hight *H*	Area *A*	Volume *V*	Weight *G*	Unit Weight *γ_avg_*
		[m]			[m^2^]	[m^3^]	[kN]	[kN/m^3^]
P-1	135	2.070	1.310	0.747	2.712	2.026	10.18	5.03
P-2	135	2.040	1.317	0.757	2.689	2.034	10.02	4.93
P-3	135	2.050	1.310	0.740	2.686	1.987	9.92	4.99
P-4	135	2.040	1.323	0.737	2.699	1.989	10.14	5.10
P-5	135	2.070	1.317	0.750	2.726	2.045	10.29	5.03
P-6	135	2.060	1.295	0.750	2.668	2.001	10.12	5.06
Avg.	135	2.055	1.312	0.747	2.697	2.014	10.11	5.02

**Table 2 materials-14-03310-t002:** Cyclic load characteristics.

Bale No.	Cyclic Load [kN]		Amplitude [kN]	Frequency [Hz]	Stress Transferred Pointwise to the Specimen [kPa]		Number of Cyclic Loads [No.]
	Max.	Min.			Max.	Min.	
P-1	20	3	17	0.4	283	42.5	10,000
P-2	20	3	17	0.4	283	42.5	10,000
P-3	20	15	5	0.8	268.9	212.3	16,000

**Table 3 materials-14-03310-t003:** Average vertical and horizontal displacements at P = 250 kN and at failure load.

Bale No.	*v_avg_ (P_250kN_)*	*h_avg_ (P_250kN_)*	*P_max_*	*v_avg_ (P_max_)*
	[mm]	[mm]	[kN]	[mm]
P-1	102	25	436	156
P-2	90	9	355	116
P-3	69	10	396	120
P-4	79	12	367	126
P-5	65	11	366	113
P-6	70	16	370	116
Avg	79	14	382	125

**Table 4 materials-14-03310-t004:** Average Young’s moduli and Poisson’s ratios for tested tyre bales.

Load Range (Cycle)	P-1 Bale		P-2 Bale		P-3 Bale		P-4 Bale		P-5 Bale		P-6 Bale	
	*E_v_*	*ν*	*E_v_*	*ν*	*E_v_*	*ν*	*E_v_*	*ν*	*E_v_*	*ν*	*E_v_*	*ν*
	kPa	[-]	kPa	[-]	kPa	[-]	kPa	[-]	kPa	[-]	kPa	[-]
0–50 kN (I)	960	0.09	620	0.06	715	0.05	690	0.06	725	0.05	720	0.10
0–50 kN (II)	1040	0.11	770	0.07	930	0.08	830	0.08	900	0.06	940	0.14
0–100 kN (I)	905	0.12	760	0.11	915	0.09	805	0.09	960	0.08	900	0.14
0–100 kN (II)	870	0.16	780	0.13	980	0.11	865	0.11	1020	0.10	965	0.15
0–250 kN (I)	590	0.19	680	0.14	860	0.11	765	0.13	890	0.13	835	0.18
0–250 kN (II)	570	0.22	670	error	730	0.11	790	0.11	930	0.16	865	0.21
Avg	822.5	0.15	713.5	0.10	855.0	0.09	791.0	0.10	904.0	0.10	871.0	0.15

**Table 5 materials-14-03310-t005:** Shear and normal stresses along the tyre bale interface.

Bale No.	Area A	Normal Load *V*	Failure Shearload *H_f_*	Normal Stress *σ*	Shear Stress *τ*	Approx. Shear Stress *τ_app_*	Error Estimation
	[m^2^]	[kN]	[kN]	[kPa]	[kPa]	[kPa]	[%]
P-1	2.712	10.00	10.53	3.69	3.88	3.85	0.84
		13.40	14.61	4.94	5.39	5.15	4.61
		20.00	21.00	7.37	7.74	7.67	0.94
		26.20	27.66	9.66	10.20	10.04	1.58
P-2	2.689	10.00	12.69	3.72	4.72	3.88	21.53
		13.40	15.29	4.98	5.69	5.19	9.49
		20.00	21.47	7.44	7.98	7.74	3.20
		26.20	30.33	9.74	11.28	10.13	11.39
P-3	2.686	10.00	8.33	3.72	3.10	3.89	20.22
		13.40	13.95	4.99	5.19	5.20	0.11
		20.00	18.70	7.45	6.96	7.75	10.11
		26.20	28.08	9.75	10.45	10.14	3.13
P-4	2.699	10.00	9.87	3.71	3.66	3.87	5.48
		13.40	13.78	4.96	5.11	5.17	1.33
		20.00	19.35	7.41	7.17	7.71	6.99
		26.20	24.12	9.71	8.94	10.09	11.42

**Table 6 materials-14-03310-t006:** Shear and normal stresses along the tyre bale–soil interface.

Bale No.	Area A	Normal Load *V*	Failure Shear Load *H_f_*	Normal Stress *σ*	Shear Stress *τ*	Approx. Shear Stress *τ_app_*	Error Estimation
	[m^2^]	[kN]	[kN]	[kPa]	[kPa]	[kPa]	[%]
P-1	2.686	22.2	14.01	8.27	5.22	5.47	4.58
		28.4	18.24	10.57	6.79	6.78	0.18
		38.4	23.00	14.30	8.56	8.90	3.74
		44.0	25.71	16.38	9.57	10.08	5.05
P-2	2.712	22.2	13.86	8.19	5.11	5.42	5.73
		28.4	18.61	10.47	6.86	6.72	2.10
		38.4	23.00	14.16	8.48	8.82	3.82
		44.0	28.03	16.22	10.34	9.99	3.45
P-3	2.726	22.2	16.45	8.14	6.03	5.40	11.80
		28.4	17.80	10.42	6.53	6.69	2.40
		38.4	24.70	14.09	9.06	8.78	3.25
		44.0	28.41	16.14	10.42	9.94	4.81

**Table 7 materials-14-03310-t007:** Creep test results.

Slab No.	*h_0_*	*t*	Δ*h_exp_*	Fitted Regression Line	*R* ^2^	Δ*h_365_*	*c_α,_* _365_
	[mm]	[h]	[mm]			[mm]	
Specimen P-1							
1	640.000	20.00	19.691	y = 0.8136 ln x + 16.454	0.9795	-----	-----
2	620.309	7.16	6.139	y = 0.2976 ln x + 5.551	0.8906	8.253	0.0034
3	614.170	19.89	8.521	y = 0.6348 ln x + 6.5415	0.9860	12.304	0.0051
4	605.649	24.11	8.967	y = 0.8372 ln x + 6.1902	0.9682	13.790	0.0058
Total		71.16	43.318	Average			0.0048
Specimen P-2							
1	665.000	20.23	21.048	y = 0.6154 ln x + 19.216	0.9569	-----	-----
2	643.952	25.07	7.774	y = 0.3026 ln x + 6.8493	0.9555	9.596	0.0038
3	636.178	24.58	7.712	y = 0.6807 ln x + 5.6076	0.9542	11.787	0.0047
4	628.466	24.79	7.333	y = 0.4696 ln x + 5.8459	0.9898	10.109	0.0041
Total		94.67	43.867	Average			0.0042
Specimen P-3							
1	666.000	21.34	8.534^(2)^	y = 0.407 ln x + 7.3927	0.6542^2)^	-----	-----
2	657.466	24.54	8.579	y = 0.6143 ln x + 6.6308	0.9406	12.207	0.0047
3	648.887	24.56	6.971	y = 0.3464 ln x + 5.7583	0.9407	8.723	0.0034
4	641.916	94.95^(1)^	8.432	y = 0.8042 ln x + 4.452	0.9489	8.813	0.0035
Total		165.39	32.516	Average			0.0039

## Data Availability

The data presented in this study are available on request from the corresponding author. The data are not publicly available due to the R&D project restrictions.

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
