# Peer review of "Experimental Determination of Mechanical Properties of Waste Tyre Bales Used for Geotechnical Applications"

_materials, 2021, doi:10.3390/ma14123310_

Round 1

Reviewer 1 Report

The authors report on a detailed experimental study, performed by laboratory mechanical tests on waste tyre bales. They showed that these waste tyre derived products (TDP) offer a useful construction mean in a wide range of geotechnical applications also thanks to their low density, high  permeability and high tyre – tyre and tyre - bale interface shear strength.

The paper sounds good; it is well written and organized. The mechanical tests are well decribed and the results well presented. 

Just few comments:

- please delete rows from 96 to 104. I guess that they are guidelines for authors, that the authors probably forget to delete.

In conclusion, the manuscript deserves publication on the journal Materials in the present form.

Reviewer 2 Report

It goes in attached document

Reviewer 3 Report

It is an experimental study that can fit into environmental geotechnics, absorbing the models of classical soil mechanics, and applying them in a model of application to the circular economy, however:

  • Is the arrangement of tires on the bale random or does it obey some preferential orientation? And this structural variation has influence on results, in compression and cyclic tests? Although they use the same type of bales.
  • In the item "Full-scale direct shear stress" the tests are used according to what type of standardization? is the shear speed constant? it would be interesting to know the influence on the results by the boundary conditions; explain better the failure criterion used (around line 350, 363, 387), as its use should be virtual, as there are conditions of different material interfaces, compressible and not confined;
  • As concludes at line 484 and 485 by friction angle difference obtained by direct shear box;
  • In all the legend of figures delete “this is a figure”. Figure 3 must be following line 141 and should classify the soil according to ASTM or British Standards or ISO (YY-axis must be: Passing weight). Figure 12 must be following line 296 and never before. Describe better the legend of figure 22 and figure 24;
  • Change the units and symbols to the international standards, e.g., Weight (kN) Unit weight (kN/m3) (Table 1), uniformizer the use of kPa or kN/m2 units, uniformizer the frictional angle symbol according to standards in eq. 6, lines 391, 395 and 396, Fig 16 and Fig 20.
  • The bibliography is adequate.

Round 2

Reviewer 2 Report

Goes in attachment

Author Response

The authors thank the Reviewer for his detailed remarks on the paper structure and some mistakes. The authors would like to respond to the Reviewer's comments as follows:

1. Abstract has been rewritten, see the part in yellow. 

2. The paper structure has been adjusted according to the Reviewer's comments and the journal's regulations. 

3. Lines 126 and 636 have been improved (superscripts). 

4. The summary has been improved and rewiriten.

 The authors hope the paper after this improvement can be published in Materials